# Exploring the Role of BCL2 Interactome in Cancer: A Protein/Residue Interaction Network Analysis

**DOI:** 10.3390/biology14030261

**Published:** 2025-03-05

**Authors:** Sidra Ilyas, Donghun Lee

**Affiliations:** Department of Herbal Pharmacology, College of Korean Medicine, Gachon University, 1342 Seongnamdaero, Sujeong-gu, Seongnam-si 13120, Republic of Korea

**Keywords:** BCL2, cancer, residue interaction, docking, MD simulations, oncogenes

## Abstract

BCL2 is a protein that plays a key role in controlling cell death, which is important for normal body functions but can also contribute to cancer. This study focused on understanding how BCL2 interacts with partner proteins and how these interactions influence cancer progression and resistance to treatments. Researchers used advanced bioinformatics tools to map out the network of proteins that bind to BCL2 and found three key partners—p53, RAF1, and MAPK1—that are involved in cancer-related processes. They also studied how these proteins interact with BCL2 at the molecular level, identifying specific novel sites on BCL2 that are critical for these interactions. Simulations showed that when p53 binds to BCL2, it weakens BCL2’s ability to prevent cell death, which could help fight cancer. On the other hand, new interactions with RAF1 and MAPK1 seem to strengthen BCL2’s cancer-promoting activity. These findings shed light on how BCL2 works in cancer and suggest that targeting its interactions with these key proteins could lead to new cancer treatments. This research provides valuable insights that could help develop therapies to better manage cancer and related diseases.

## 1. Introduction

The B-cell lymphoma 2 (BCL2) protein family, located on human chromosome 18, plays a crucial role in regulating cell death by apoptosis, a process essential for maintaining cellular homeostasis and preventing cancerous transformations. This family includes pro-survival and anti-apoptotic proteins that control the mitochondrial outer membrane (MOMP), which is responsible for releasing cytochrome c, a critical event in apoptosis [1,2]. Dysregulation of apoptotic regulators is frequently observed in various cancers, contributing to uncontrolled cell growth, differentiation, survival, and resistance to therapeutic treatments. BCL2 is often overexpressed during tumor development, indicating its importance as a survival factor. It inhibits intrinsic apoptosis pathways by controlling the mitochondrial membrane’s permeability, inhibiting the release of cytochrome c, or by preventing caspase activation by binding to apoptosis-inducing factor (AIF-1) [1]. Additionally, BCL2 reduces inflammation by impairing NLRP1 inflammasome activation, leading to reduced CASP1 activation and IL1B production, which contribute to modulating the immune responses [3].

Apoptotic regulators can be categorized into three groups: pro-survival (BCL2-like proteins), pro-apoptotic BH3-only proteins, and pro-apoptotic effector proteins. BCL2 proteins contain at least one or all four conserved homology domains (BH1-4) and display two central hydrophobic α-helices surrounded by six or seven amphipathic α-helices of varying lengths [4]. Pro-survival members such as BCL2, BCL-XL, BCLW, MCL-1, and A1 have a hydrophobic groove on their surface, which facilitates binding to various pro-apoptotic proteins such as BAX, BAK, and BID [5,6,7,8]. Only the BH3 region of pro-apoptotic proteins interacts with the pro-survival members. The multi-domain homologs such as BAK and BAX promote apoptosis, whereas others such as BCL2 and BCL-XL protect against apoptosis [2]. The BH3-only protein contains a single domain responsible for interacting and regulating the function of the multi-domain homologs. BH3 members are pro-apoptotic in their behavior and are responsible for controlling the system that promotes the pro-apoptotic effects of BAK and BAX [9]. Different binding affinities to multi-domain proteins have been observed in BH3-only proteins. This feature was associated with the sequences exhibited in the surface groove. BCL2 binding to BAX blocks the release of cytochrome c from the mitochondria.

BCL2 regulates apoptosis through interactions with various proteins. It forms homo/heterodimers with pro-apoptotic proteins (BAX, BAD, BAK) being essential for its anti-apoptotic function [10]. It forms a complex with proteins such as EI24, APAF1, BBC3, TP53BP2, and FKBP8 modulating its apoptotic functions [3]. Other interactions, including with BAG1, RAF1, and EGLN3, further regulate its anti-apoptotic activity by disrupting the BAX-BCL2 complex. These diverse interactions highlight BCL2’s role in balancing apoptosis and autophagy [11].

Cellular survival is governed by signaling pathways such as PI3K/AKT, JAK-STAT, and ERK1/2. AKT activation in the PI3K pathway phosphorylates and inhibits BCL2 family members promoting cell survival. Meanwhile, the JAK/STAT pathway induces BCL2 proteins and ERK1/2 signaling that enhances the transcription of BCL2 and BCL-XL through CREB phosphorylation. This intricate interplay between these signaling pathways and the BCL2 protein family ensures the regulation of apoptosis and cellular integrity. As the BCL2 protein family is involved in various signaling pathways, its dysregulation facilitates uncontrolled cell proliferation and contributes to therapy resistance.

The aim of this study is to explore the protein–protein interaction (PPI) of BCL2 focusing on cancer-related proteins and their potential as a therapeutic target. To achieve this goal, a PPI network using PINA platform was generated to identify key interactors of BCL2 within the context of cancer biology. MOE, STRING, and gProfiler were utilized to investigate the interactions, and functional and biological annotations. Docking studies were performed to elucidate the novel interactions due to unavailable experimental structures of BCL2-MAPK1, and BCL2-RAF1. MD simulations (200 ns) of the key identified proteins were conducted to analyze the stability and conformational changes of protein complexes over time. By integrating the multi-dimensional computational approaches, a deeper insight into the molecular mechanisms of BCL2-associated partner proteins and their role in cancer can be identified.

## 2. Materials and Methods

The workflow of this study is presented in Figure 1. The methodology consisted of data compilation, data cleaning, protein–protein interaction (PPI) analysis, hub genes identification, molecular docking, and MD simulation.

### 2.1. Data Collection and Cleaning

The protein–protein interaction (PPI) network of BCL2 was generated by the PINA (v3.0) “URL https://omics.bjcancer.org/pina/queryProteinSet.action platform (accessed on 7 November 2024)”. The initial query for BCL2 (Uniprot: P10415) yielded 133 unique protein interactors retrieved from all sources and large-scale studies available in the PINA repository. The PPI dataset was filtered to include only direct interactions, focusing on an experimentally validated interactome. This filtering further reduced the total number of interactions from 133 to 59 direct interactors. Direct interactions were defined as those where direct molecular binding/association between two proteins had been confirmed by experimentally verified sources. Duplicate interactors were removed to create a non-redundant list.

### 2.2. Cancer Drivers and Drug Target Selection

The list of direct interactors was further filtered to focus on proteins relevant to cancer to include only those known to be cancer drivers and cancer drug targets. Cancer drivers and cancer drug targets were selected, and the common genes were identified. Subsequently, the interactors for each of the identified genes were queried using PINA. The results for each gene yielded a number of interactors that were filtered based on publication references. Data were cleaned to remove redundancy ensuring that each gene was represented only once.

### 2.3. Identification of Common Interactors with BCL2

VENNY (v2.1) “URL https://bioinfogp.cnb.csic.es/tools/venny/index.html (accessed on 9 October 2024)” was used to identify common interactors between the BCL2 network and key interactors. The intersection analysis revealed a set of common genes, which was used for further functional annotation to explore their roles in cancer.

### 2.4. PPI Network Using STRING

The complex relationships between proteins and functional associations were analyzed by STRING “URL https://string-db.org/ (accessed on 9 October 2024)” and species confined to “Homo sapiens”. STRING provides a comprehensive collection of known and predicted PPIs, integrating both experimental data and computational predictions. The interaction network was generated by selecting a confidence threshold (0.7) to filter reliable interactions. The resulting PPI network was further explored for hub genes and functional enrichment analysis.

### 2.5. Identification of Hub Genes by Using Cytoscape

Hub genes in the BCL2 network were identified by using the Cytoscape plugin 3.10.0 (cytoHubba) based on centrality metrics [12]. Degree centrality and shortest path were used for identification of hub genes. Genes are ranked according to the number of direct interactions (edges) in the network; those with the highest degree are referred to as hub genes since they are highly connected. Conversely, the shortest path approach determines the bare minimum of steps needed to connect genes throughout the network, emphasizing genes that might not be directly associated but are essential for connecting various network parts.

### 2.6. Functional Enrichment Analysis

To gain insights into the biological processes (BPs), cellular components (CCs), and molecular functions (MFs) associated with the interactors, functional enrichment analysis was performed using gProfiler “URL https://biit.cs.ut.ee/gprofiler/gost (accessed on 19 October 2024)”. Additionally, Kyoto Encyclopedia of Genes and Genomes (KEGG) pathway enrichment analysis was performed. These tools were used to identify overrepresented pathways and cellular functions associated with the genes.

### 2.7. Protein Structures’ Retrieval and Preparation

The three key proteins (p53, RAF1, and MAPK1) associated with BCL2 were selected for docking studies. The 3D protein structures were obtained from the Protein Data Bank (PDB) “URL https://www.rcsb.org/ (accessed on 25 November 2024)”. The PDB IDs of BCL2-p53 (8HLM), BCL2-RAF1 (3OMV), and BCL2-MAPK1 (2GPH) were utilized. The structures were optimized by removing water molecules and any heteroatoms that could interfere with docking. While the crystal structure of BCL2-p53 (8HLM) was available, for BCL2-RAF1 and BCL2-MAPK1, experimental structures were unavailable; hence, docking was essential to predict their binding modes, key residues, and interaction dynamics. These PDB IDs (3OMV and 2GPH) correspond to structures of RAF1 and MAPK1, respectively, in a complex with other proteins/compounds, not with BCL2. Therefore, we used the crystal structures of RAF1 (3OMV) and MAPK1 (2GPH) as templates for docking simulations. These docking studies provided a unified framework for analyzing the binding characteristics and potential functional implications of these complexes.

### 2.8. Protein–Protein Docking

Docking was performed by using the HDOCK server “URL http://hdock.phys.hust.edu.cn/ (accessed on 28 November 2024)”, which integrates both template-based and ab initio methods. Each of the three complexes (BCL2-p53, BCL2-RAF1, and BCL2-MAPK1) was studied separately by submitting the prepared protein structures to the HDOCK server. For each protein pair, the server produced docking poses based on the predicted interaction energies, with the lowest energy conformations considered as the most probable binding states. Interacting residues at the protein interface were identified using the RING server “URL https://ring.biocomputingup.it (accessed on 3 December 2024)”, which highlights key amino acids involved in the interaction. The interaction data were visualized using a chord diagram developed by raw graphs “URL https://www.rawgraphs.io/ (accessed on 6 December 2024)” illustrating the connectivity and distribution of interacting residues between the proteins. The docking interaction poses were visualized using PyMOL “URL https://www.pymol.org/ (accessed on 11 December 2024)”, highlighting key connections including hydrogen bonds and hydrophobic interactions.

### 2.9. Contact Analysis

The contact analysis feature in MOE (v2020.09) “URL https://www.chemcomp.com/en/Products.htm (accessed on 16 December 2024)” was employed to examine protein–protein interactions, identifying six types of contacts: hydrogen bonds, and metal, ionic, covalent, Arene, and van der Waals distance interactions. The analysis involved calculating contact surfaces as sets of points equidistant between two proteins within a specified minimum distance. The Generalized Born with Volume Integral (GBVI) method was used to estimate the total binding affinity for each protein–protein complex.

### 2.10. Molecular Dynamics (MD) Simulation

MD simulations (200 ns) were performed using the Gromacs 2024.1 program “URL https://www.gromacs.org/ (accessed on 20 December 2024)” to investigate protein–protein interactions. The complex was modelled using the CHARMM36 all-atoms force field and embedded using the TIP3p water model in a cubic box with a 12 Å buffer distance [13]. The system was neutralized with counter ions such as sodium (Na^+^) and chloride (Cl^−^) to mimic physiological conditions. Energy minimization was conducted using the steepest descent algorithm until the system reached a gradient tolerance of 0.001 kJ/mol. To achieve equilibration, the systems were subjected to temperature and pressure control using the Nose-Hoover and Parrinello–Rahman methods, respectively [14]. Short-range electrostatic and van der Waals interactions were calculated within a cutoff radius of 1.2 nm, while long-range electrostatic interactions were treated using the particle mesh Ewald method [15]. Post-simulation analyses were performed to assess the stability, conformational dynamics, and interaction interface of the protein–protein complex, with visualization and trajectory conducted using Gromacs.

## 3. Results

### 3.1. Identification of Cancer Drivers

The BCL2 (Uniprot: P10415) queried in the PINA platform retrieved 133 interactors, out of which 116 interactors were experimentally validated (Appendix A). Based on the unique direct 59 interactors, cancer drivers and cancer targets were identified by a filtering method (Appendix A). Intersection analysis of cancer drivers (6) and cancer targets (9) revealed three key partners, p53, RAF1, and MAPK1, involved in cancer (Table 1). The interactome of each gene was retrieved from PINA. P53 showed 1196 interactors, of which 1156 were experimentally verified, and after removing duplicates, 1149 unique data remained (Appendix A). RAF1 revealed 315 interactors, of which 296 were experimentally verified, and after removing duplicates, 292 unique interactors remained (Appendix A). Similarly, MAPK1 showed 415 interactors in total, out of which 339 were experimentally verified, and all of which remained after duplication removal (Appendix A).

### 3.2. Common Interactors of BCL2

Eleven (11) common interactors of BCL2, p53, RAF1, and MAPK1 were identified (Figure 2). The common proteins were named as androgen receptor (AR), proto-oncogene tyrosine-protein kinase (SRC), tubulin gamma-1 chain (TUBG1), mitogen-activated protein kinase (MAPK3), prohibitin (PHB), mitogen-activated protein kinase kinase 3 (MAP2K3), valosin-containing protein (VCP), mitogen-activated protein kinase kinase kinase 1 (MAP3K1), protein phosphatase 2 regulatory subunit (PPP2R5C), tripartite motif-containing protein 25 (TRIM25), and EGL nine homolog 3 (EGLN3), which are involved in cellular processes such as signaling, apoptosis, and immune responses, indicating their potential roles in cancer (Appendix A).

### 3.3. Protein–Protein Interaction Analysis

A complex and intricate network of BCL2 interaction was discovered by using STRING (Figure 3A). Additionally, the top 10 hub genes *p53* (28), *MAPK3* (20), *MAP3K1* (18), *AR* (18), *MAPK1* (18), *SRC* (18), *RAF1* (18), *BCL2* (16), *PPP2R5C* (12), and *TRIM25* (8) were identified based on the degree centrality method and shortest path by cytoHubba (Figure 3B). These genes are highly interconnected and play central roles in key biological processes.

### 3.4. Functional Enrichment Analysis

The functional annotation of genes revealed several enriched biological processes (BPs), cellular components (CCs), and molecular functions (MFs), highlighting key pathways and regulatory mechanisms (Figure 4). The analysis identified significant enrichment in the positive regulation of molecular functions, the ERBB2 signaling pathway, MAPK cascade, positive regulation of the phosphate metabolic process, and the apoptotic pathway. These biological processes suggest a broad involvement in cellular signaling, metabolism, and transcriptional regulation. In terms of CCs, genes were significantly enriched in the cytosol, nucleoplasm, caveola, and endosome, indicating a diverse range of cellular localization, with a prominent presence in both the cytoplasm and nucleus. This distribution underlines the importance of intracellular signaling and gene expression regulation. At the MFs level, the identified genes exhibited substantial enrichment in enzyme binding, phosphoprotein binding, heterocyclic compound binding, and MAP protein kinase activity. These molecular functions suggest active involvement in enzyme catalysis, molecular interactions, and signal transduction, particularly through protein phosphorylation, which is consistent with the regulatory roles of MAPK and steroid receptor signaling pathways (Figure 4). The KEGG pathway enrichment analysis revealed several significantly enriched pathways, with particular emphasis on the Hepatitis B, GnRH signaling pathway, EGFR inhibitor resistance, endocrine resistance, and prostate cancer. These pathways are crucial for signaling, resistance, and cancer.

### 3.5. Docking Studies

The docking analysis revealed several significant hydrogen bonds and van der Waals interactions between p53 and BCL2, highlighting key amino acid pairs involved in the binding interface (Appendix A). Notably, the hydrogen bond interactions include p53-SER99 with BCL2-ASP111, p53-ASN131 with BCL2-ASP140, and p53-SER166 with multiple residues of BCL2 such as ALA100, ASP103, and ARG107 (Figure 5). Additionally, p53-SER269 forms hydrogen bonds with both ASP140 and ARG146 of BCL2. On the other hand, van der Waals interactions were observed between several amino acids, including p53-VAL97 with BCL2-ARG107, p53-SER99 with BCL2-TYR108, and p53-ASN268 with BCL2-ASP140 (Figure 6). These interactions suggest a complex network of residue pairings contributing to the structural stability of the BCL2-p53 complex (Appendix A).

The interaction analysis between RAF1 and BCL2 revealed multiple hydrogen bonds, π-π stacking interactions, and van der Waals interactions, suggesting a complex and stable binding interface (Figure 7). Key hydrogen bonds were observed between LYS470, ASN472 of RAF1 with ASP103, ARG107 of BCL2, and RAF1-LYS431 with both TYR108 and ASP111 of BCL2. Additionally, interactions between RAF1-GLY361 and BCL2-ARG109, and between RAF1-ASP555 and BCL2-LEU201, further contributed to the stability of the complex. The π-π interactions observed were between RAF1-TYR430 with BCL2-PHE104 and TYR108 as well as RAF1-TYR548 and BCL2-TRP144 (Figure 8). The van der Waals interactions included pairs such as RAF1-VAL509 with BCL2-GLN99, and RAF1-LYS431 with ARG107 and ARG110 of BCL2 (Appendix A). These diverse interactions indicate a multifaceted interaction between RAF1 and BCL2, supporting a strong binding interface (Appendix A).

The interaction analysis between MAPK1 and BCL2 identified several significant hydrogen bonds, π-π stacking, π-cation, ionic interactions, and van der Waals contacts (Appendix A). Key hydrogen bonds observed were between residues such as ARG13, TYR28, and LYS115 of MAPK1 with ARG107, GLU136, and TRP144 of BCL2 (Figure 9). Additionally, an ionic interaction was identified between MAPK1-LYS115 with BCL2-GLU136. Van der Waals interactions were prevalent, involving multiple residues on both proteins. Finally, π-cation interactions were detected between TYR111, TYR28, and TYR185 of MAPK1 with ARG146, TYR108, and TRP144 of BCL2, respectively (Figure 10). The docking and confidence scores for BCL2 interactions with p53, RAF1, and MAPK1 by HDOCK server are shown in Appendix A.

### 3.6. Contact Analysis

The 3D crystal structure of BCL2 with associated key partner proteins involved in apoptosis regulation were retrieved from PDB (Table 2). Contact analysis of the BCL2 protein family revealed a diverse network of interactions, suggesting that BCL2 acts as a central regulator of apoptosis, inhibiting cell death by sequestering pro-apoptotic proteins. The formation of homodimers and heterodimers within BCL2 further highlights the complexity of the regulatory mechanisms involved (Figure 11). The results showed that hydrogen and ionic bonds made by ASP101 and ARG128 residues are dominant among the structure complexes, with the maximum interaction energy being –24.85 Kcal/mol (Appendix A). The least favorable energy is due to the ionic–hydrogen interaction between ASP138 and ARG105 residues. Ionic and hydrogen interactions are prominent at more than 2.6 Å distance (Appendix A).

### 3.7. Molecular Dynamics Studies

A lower RMSD indicates a more stable and rigid structure in contrast to a higher RMSD suggesting flexibility and conformational changes. Both proteins exhibited an increase in RMSD upon BCL2-p53 complex formation, suggesting more flexible and dynamic behavior (Figure 11). The complex exhibited a rapid increase in RMSD during the initial phase, followed by a plateau phase of the simulation. This increased flexibility could also lead to misfolding and/or aggregation of BCL2, which could negatively affect anti-apoptotic activity. The RMSF analysis of each protein revealed distinct patterns of flexibility with p53 DNA-binding domain (DBD) showing higher RMSF values in the beginning at the N-terminal region (residues 95–100), suggesting that this domain is quite flexible compared to other regions of the protein. BCL2 (BH3 pocket) also exhibited higher RMSF values, particularly in the N-terminal region (residues 1–50), indicating increased flexibility. The radius of gyration (Rg) fluctuated around an average value of about 2.3 nm, indicating a relatively stable overall conformation. The hydrogen bond analysis revealed the number of hydrogen bonds fluctuated over time, with an average of approximately 6–8 bonds present at any given time point. Additionally, several specific hydrogen bonds formed and broke repeatedly during the simulation, highlighting their potential role in the protein’s functional state (Figure 12).

The RMSD analysis of the BCL2-RAF1 complex and its individual proteins revealed distinct dynamic behavior. The complex exhibited a rapid increase in RMSD during the initial phase, followed by a plateau phase after 90 ns (Figure 12). The initial rise reflects the system equilibration to the simulation conditions. The subsequent plateau suggests that the complex reaches a relatively stable conformation. In contrast, the RAF1 protein exhibited a gradual increase in RMSD over time indicating greater flexibility compared to BCL2, which exhibited lower flexibility. BCL2 exhibited a lower RMSF, indicating rigidity and stability over RAF1, which showed a higher RMSF in the region between residues 300 and 600, suggesting flexibility and conformational changes. The Rg fluctuated around an average value of approximately 2.3 nm, indicating a stable overall conformation. The hydrogen bonding interactions between proteins fluctuated over time with an average of approximately 6–8 bonds present at a given period, indicating hydrogen bonds provide stability to the proteins (Figure 13).

The RMSD plot of the BCL2-MAPK1 complex showed a sharp rise in RMSD during the simulation’s first phase, followed by a plateau phase. This initial increase indicates that the system has adjusted to the simulation’s settings. The complex appears to achieve a stable shape at the plateau phase. For the BCL2-MAPK1 complex, the RMSD increased from 0.4 to 0.6 nm between 40 and 60 ns, before decreasing to approximately 0.5 nm. However, both individual proteins showed a steady RMSD over time, suggesting a comparatively stable structure. The RMSF analysis showed distinct patterns of flexibility. MAPK1 showed a comparatively constant degree of fluctuation throughout the simulation. Nonetheless, there were noticeable peaks in the RMSF, especially close to the C-terminus and around residue 150. In contrast, BCL2 showed increased fluctuations in specific regions, notably around residues 50–100 and 300–350. These regions likely correspond to loops or domains that are more flexible in BCL2 compared to MAPK1. The Rg fluctuated around an average value of approximately 2.45 nm, indicating a relatively stable overall conformation. The hydrogen bond analysis revealed dynamic interactions throughout the simulation with an average of approximately 8–10 hydrogen bonds present at any given time point (Figure 14).

## 4. Discussion

In this study, a known cancer target, BCL2, in complex with partner proteins was explored to highlight the multifaceted role of BCL2 in regulating cell survival and cell death. An intricate network of the BCL2 protein family tightly regulates apoptosis. The contact analysis provides valuable insights into this intricate network, crucial for regulating apoptosis, development, tissue homeostasis, and disease. Understanding these interactions is crucial for developing therapeutic strategies targeting apoptosis, such as cancer and neurodegenerative diseases. BCL2 regulates apoptosis and cell survival by forming homo/heterodimers with partner proteins. For instance, its heterodimerization with BAX requires BH1 and BH2 motifs, which are essential for its anti-apoptotic activity [39]. The BCL2 complex with XIAP, and ARTS (a scaffold protein), can trigger apoptosis [28,29]. Under non-starvation conditions, BCL2 interacts with BECN1 to prevent the formation of an autophagy-inducing complex with PIK3C3 [40]. Furthermore, it binds to APAF1, BBC3, BCL2L1, BNIPL, and p53 [3]. p53, a well-known tumor suppressor, regulates apoptosis and cell cycle checkpoints in response to DNA damage and stress. It can directly interact with BCL2, disrupting BCL2’s anti-apoptotic function to promote apoptosis leading to cell death [41]. Other significant interactions include those with BAG1, RAF1, and EGLN3, which could also modulate BCL2 anti-apoptotic activity [3]. RAF1 can indirectly modulate BCL2 activity through the MAPK signaling cascade. RAF1 activates MAPK1 through phosphorylation, and regulates downstream targets that can promote survival. By phosphorylating BAD at SER-75 position, RAF1 translocates to the mitochondria, where it binds BCL2 and displaces BAD, further protecting cells from mitochondrial-mediated cell death and regulating apoptotic signaling [42,43,44].

The results from STRING and cytoHubba highlight a set of genes that are likely to play pivotal roles in cell fate decisions like proliferation, differentiation, and apoptosis. The identification of *p53*, *MAPK3*, *MAP3K1*, and other kinases as central hubs in the network emphasizes their importance in maintaining cellular integrity and regulating critical pathways. The involvement of *BCL2* and *AR* within this network suggests that key survival and differentiation mechanisms are also tightly integrated, particularly in cancerous states where dysregulation of these pathways is common [45]. The high degree of interaction observed in genes like *MAPK3*, *MAP3K1*, *SRC*, and *RAF1* further supports their roles in regulating cellular signaling, which is crucial for understanding cancer biology and therapeutic resistance [46]. *RAF1* and *MAPK1* (also known as *ERK2*) are central components of the MAPK/ERK signaling pathway, which regulates key cellular processes, including cell growth, differentiation, and survival. *PPP2R5C* and *TRIM25*, although less connected than the top genes, may play modulatory roles in regulating phosphorylation and immune signaling pathways for maintaining cellular homeostasis, and may offer insights into novel therapeutic approaches, particularly in diseases associated with immune dysregulation or protein misfolding.

The functional annotation of genes revealed significant enrichment in various biological processes, cellular components, and molecular functions, shedding light on critical regulatory mechanisms within the cell. Enrichment in processes such as the positive regulation of molecular functions, the ERBB2 signaling pathway, MAPK cascade, phosphate metabolism, and apoptosis points to a broad involvement in cellular signaling, metabolic regulation, and transcriptional control. The localization of these genes to key cellular components, including the cytosol, nucleoplasm, caveola, and endosome, emphasizes their crucial roles in both intracellular signaling and gene expression regulation. Moreover, the molecular function enrichment, particularly in enzyme binding, phosphoprotein binding, heterocyclic compound binding, and MAP protein kinase activity, underscores their involvement in catalytic processes, protein interactions, and signal transduction, especially through phosphorylation mechanisms. The KEGG pathway enrichment analysis further highlighted several critical pathways, such as Hepatitis B, GnRH signaling, and prostate cancer, which are pivotal in cellular signaling, drug resistance, and cancer progression. These findings suggest that the identified genes play a central role in modulating cellular functions and may serve as potential targets for therapeutic interventions in diseases like cancer and resistance syndromes.

Docking across all three protein complexes showed several key amino acids consistently appear, highlighting their central role in hydrogen bonds, van der Waals, π-π stacking, and π-cation in stabilizing these interactions. Residues of BCL2 such as ASP111, ASP140, ARG107, and ARG146 are involved in multiple interactions across all three complexes. For instance, BCL2-ASP111 forms multiple hydrogen bonds, and van der Waals contacts with p53-SER99, p53-SER166, and RAF1-LYS470, highlighting its central role in mediating protein–protein interaction. Similarly, BCL2-ASP140 and BCL2-Arg107 form hydrogen bonds with p53-ASN131, p53-SER269, and MAPK1-ASN472, which contribute to the structural integrity of the complex. The residue BCL2-ARG107 also forms several key interactions with p53-SER166 and MAPK1-LYS431, further emphasizing its role in maintaining the stability of the complex. Additionally, BCL2-TRP144 and TYR108 engage in π-π stacking with p53-TYR185 and RAF1-TYR430. These findings play critical roles in the protein interface, making them important targets for therapeutic interventions aimed at modulating BCL2 functions in cancer and apoptosis-related disorders. The consistency of these interactions across different complexes suggest that conserved contact points could be exploited in drug design and other applications.

The RMSD analysis of the BCL2-p53 complex and its individual proteins revealed distinct dynamic behaviors. This initial rise in RMSD indicates conformational changes in BCL2 (BH3 pocket) and p53 DNA-binding domain (DBD), and suggest that p53 is altering the structure of BCL2 upon binding. p53’s ability to inhibit BCL2 can induce apoptosis and may depend on this structural flexibility. These results highlight the differential dynamic properties of the two proteins, adjusting their conformation upon interaction. The analysis of RMSF (total fluctuation of individual amino acids over time) provides key insights into the flexible nature of p53 and BCL2. The fluctuations of p53 residues (95–100) and of BCL2 (1–50) may contribute to molecular motions. The manner in which p53 inhibits BCL2 and stops its regular function may be largely due to this dynamic flexibility and structural rearrangements. Small fluctuations in the Rg suggest that the protein undergoes subtle conformational changes during the simulation. However, no significant deviations from the average Rg were observed, indicating that the protein maintains its overall structural integrity. The BCL2-p53 complex showed a stronger hydrogen bonding, indicating a higher level of protein stability. These fluctuations in hydrogen bond formation and breakage correlate with changes in the protein’s conformation, suggesting that hydrogen bonds are integral to maintaining the stability, and play a role in folding of the protein.

The RMSD analysis of the BCL2-RAF1 complex and its individual components revealed distinct dynamic behaviors which appear to influence their functions. RAF1 exhibits increased flexibility upon complex formation, suggesting that it may undergo conformational changes. BCL2 with a lower RMSF might be involved in structural or scaffolding functions, requiring more rigid and stable conformation compared to the higher RMSF in raf1 where flexibility is involved in functional interaction. Identifying a secondary structure and functional domains could provide insights into the relationship between flexibility and function. The stable Rg suggests that protein is well folded and small fluctuations in the Rg are likely due to the thermal motions and internal dynamics of the protein to explore more conformational space. The analysis of hydrogen bond dynamics provides information about the mechanism underlying protein function, stability, and misfolding.

The RMSD plot of the BCL2-MAPK1 complex revealed a sharp initial increase followed by a plateau, indicating that the system adjusted to the simulation conditions before stabilizing. The RMSF analysis highlighted MAPK1, showing constant fluctuations near its C-terminus and around residue 150, while BCL2 exhibited greater fluctuations, particularly in specific residues 50–100 and 300–350, representing more flexible loops or domains. The Rg analysis indicated a stable overall conformation with fluctuations around an average of 2.45 nm. These results offer valuable insights into the molecular mechanisms by which BCL2 regulates apoptosis and highlight its potential as a therapeutic target in cancer, where restoring apoptotic pathways through its interaction with partner proteins may overcome therapeutic resistance and inhibit cancer growth.

## 5. Conclusions

This study illustrates a detailed investigation into the protein–protein interactions (PPIs) of BCL2 and its potential role in cancer progression and therapeutic resistance. Through the construction of a comprehensive PPI network and subsequent analysis using multiple bioinformatics tools (PINA, MOE, STRING, RING, and gProfiler), three key interactors (p53, RAF1, and MAPK1) were identified as crucial players in the cancer-related activity of BCL2. Molecular docking studies confirmed that BCL2 interacts with these proteins through novel key hydrogen bonds, ionic interactions, and van der Waals forces, contributing to the stability of the complexes. Additionally, MD simulations indicated that the binding of p53 to BCL2 resulted in a slight increase in RMSD, suggesting a potential suppression of BCL2’s anti-apoptotic activity. The RAF1-BCL2 complex also showed an increased RMSD, indicating that RAF1 enhances BCL2 activity. The BCL2-MAPK1 complex achieves structural stability over time, with regions of increased flexibility in both proteins, suggesting functional relevance in their interactions. These findings provide valuable insights into how BCL2 regulates apoptosis and emphasize its potential as a therapeutic target in cancer.

## Figures and Tables

**Figure 1 biology-14-00261-f001:**
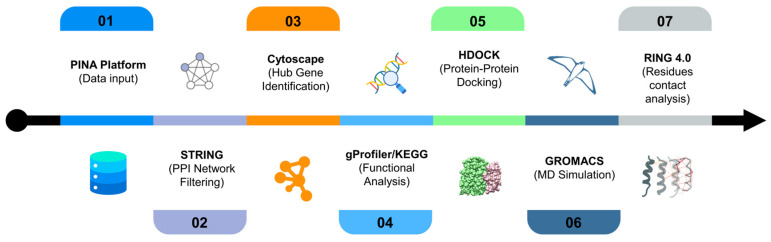
The workflow used in the study.

**Figure 2 biology-14-00261-f002:**
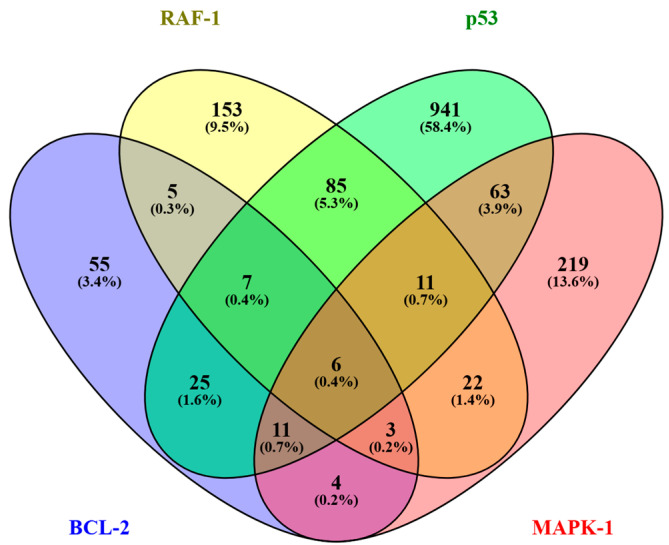
Venn diagram showing common interactors (11) of BCL2, p53, RAF1, and MAPK1.

**Figure 3 biology-14-00261-f003:**
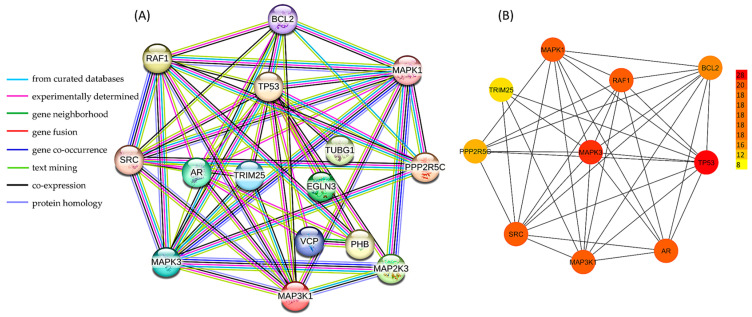
Protein–protein interaction analysis using (**A**) STRING and (**B**) cytoHubba. The STRING network was supported by multiple lines of evidence, including experimentally validated interactions and associations, further strengthening the reliability of the identified protein–protein relationships. The top 10 PPI interactions identified by cytoHubba based on degree centrality and shortest path were also identified.

**Figure 4 biology-14-00261-f004:**
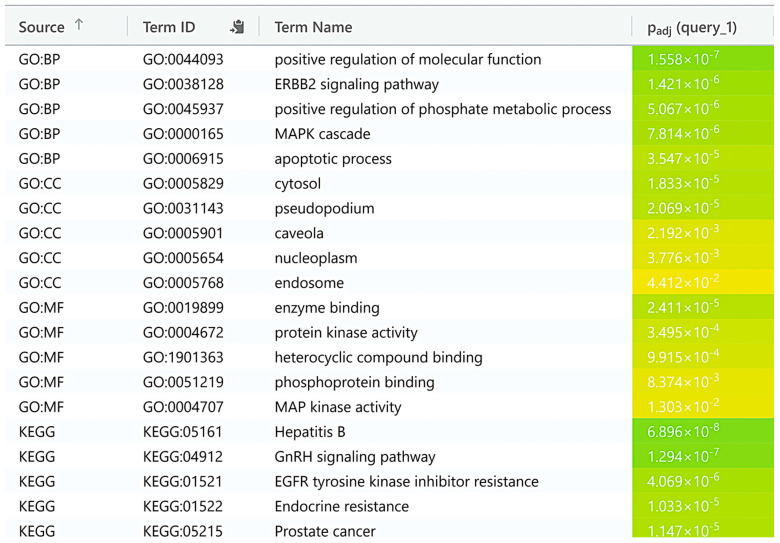
Key biological processes, molecular functions, cellular components, and KEGG pathways identified in functional enrichment analysis by gProfiler.

**Figure 5 biology-14-00261-f005:**
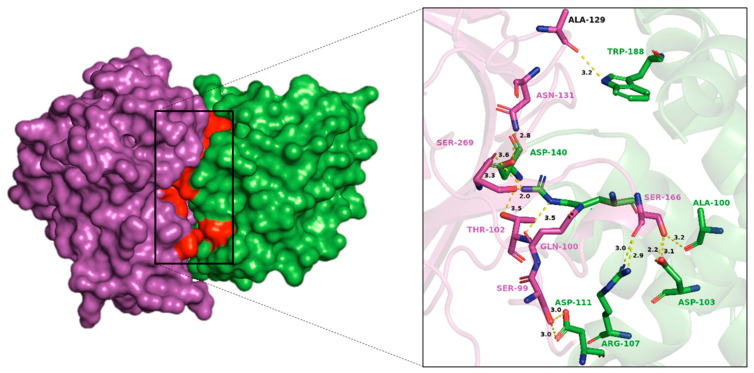
The BCL2-p53 complex generated using HDOCK server was visualized using PyMOL, highlighting key connections including hydrogen bonds and hydrophobic interactions where green shows BCL2 protein and magenta color indicates p53.

**Figure 6 biology-14-00261-f006:**
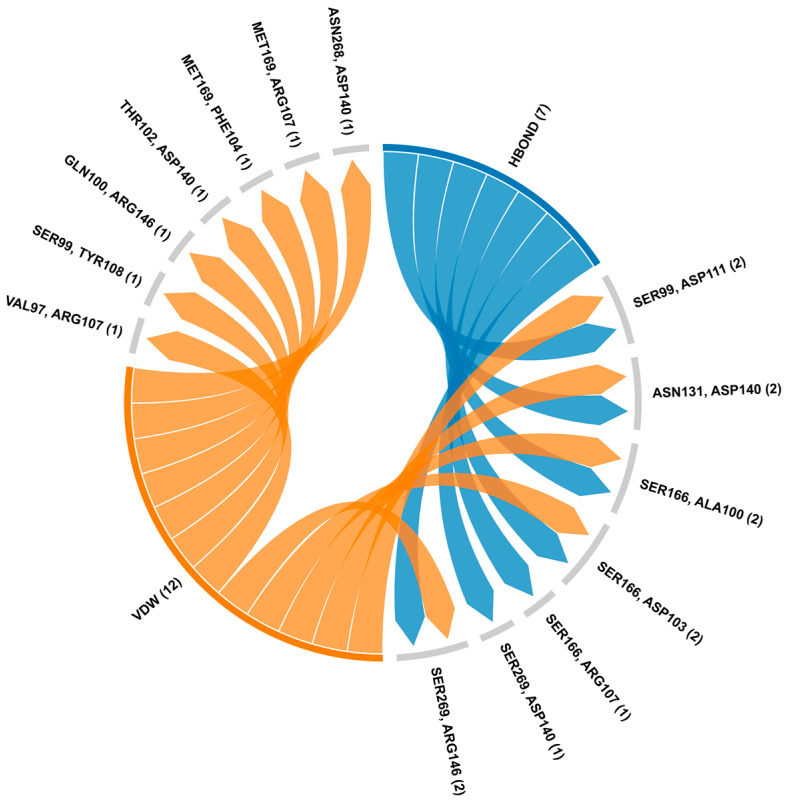
Protein–protein interaction analysis of BCL2-p53 using HDOCK and RING server. The chord diagram illustrates the connectivity and distribution of interacting residues between BCL2 and p53. The first amino acid (SER99 with ASP111) showed p53, while the second residue displayed BCL2. The HBOND indicates hydrogen bonds and VDW specifies van der Waals forces.

**Figure 7 biology-14-00261-f007:**
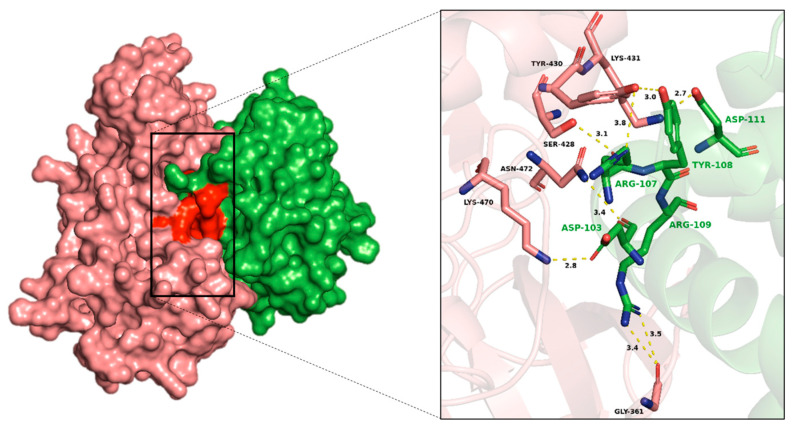
The BCL2-RAF1 complex generated using HDOCK server was visualized using PyMOL, highlighting key connections including hydrogen bonds and hydrophobic interactions where green shows BCL2 protein and salmon indicates RAF1.

**Figure 8 biology-14-00261-f008:**
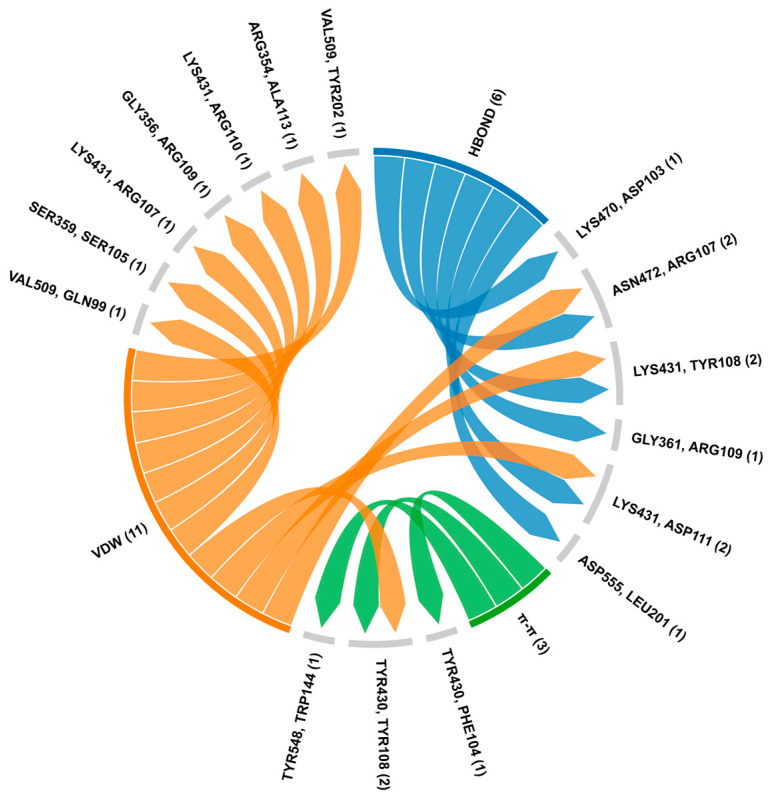
Protein–protein interaction analysis of BCL2-RAF1 using HDOCK and RING server. The chord diagram illustrates the connectivity and distribution of interacting residues between RAF1 and BCL2 where first amino acid (LYS470 with ASP103) showed RAF1 and second residue exhibited BCL2. The HBOND indicates hydrogen bonds and VDW specifies van der Waals forces.

**Figure 9 biology-14-00261-f009:**
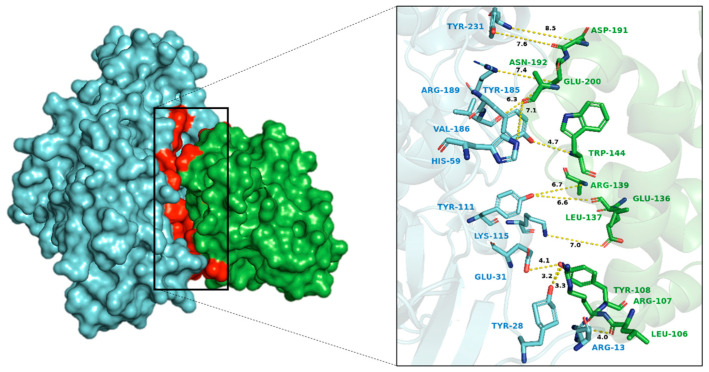
The BCL2-MAPK1 complex generated using HDOCK server was visualized using PyMOL, highlighting key connections including hydrogen bonds and hydrophobic interactions, where green shows BCL2 protein and cyan color indicates MAPK1.

**Figure 10 biology-14-00261-f010:**
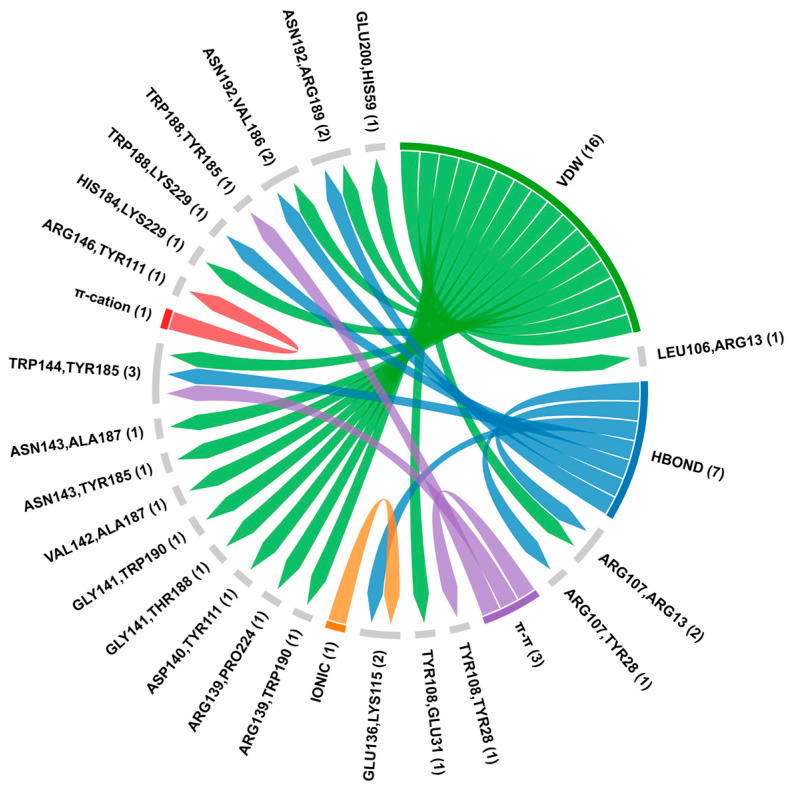
Protein–protein interaction analysis of BCL2-MAPK1 using HDOCK and RING server. The chord diagram demonstrates the connectivity and distribution of interacting residues between the proteins where first amino acid (ARG107 with ARG13) showed BCL2 and second residue displayed MAPK1. The HBOND indicates hydrogen bonds and VDW specifies van der Waals forces.

**Figure 11 biology-14-00261-f011:**
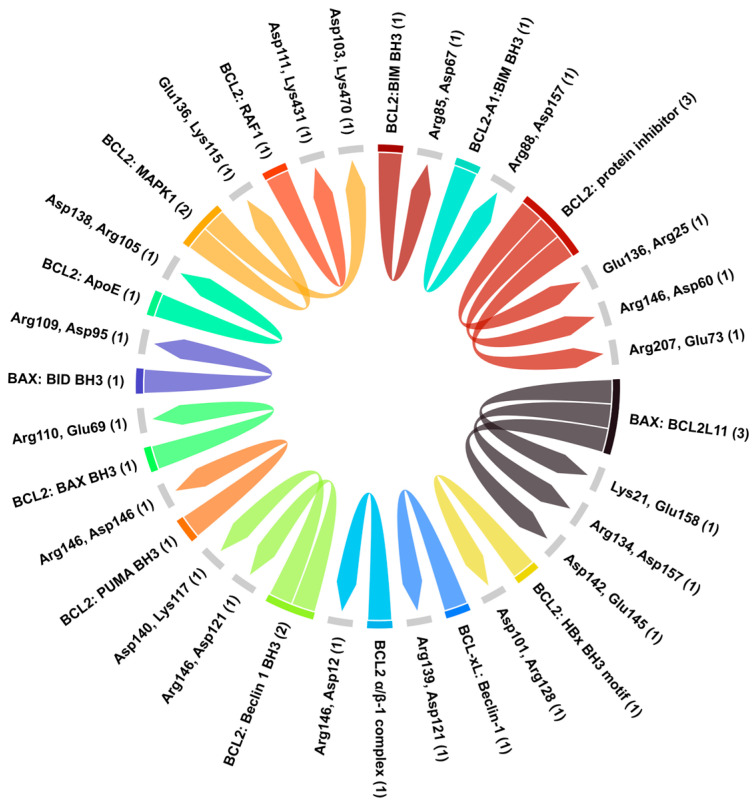
Protein–protein interaction analysis of BCL2 with associated partners. The chord diagram highlights the key amino acid residues involved in both hydrogen as well as ionic bonding.

**Figure 12 biology-14-00261-f012:**
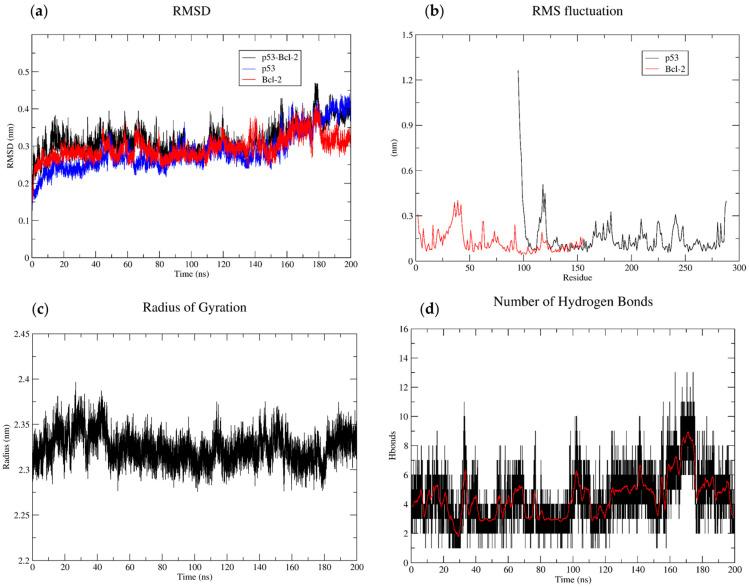
MD simulations (200 ns) of BCL2, p53, and BCL2-p53 complex. (**a**) The RMSD plot showed the stability of the complex during the simulation, with fluctuations indicating conformational changes. (**b**) The RMSF analysis highlights the flexibility of individual residues within the complex. (**c**) The radius of gyration (Rg) indicates a stable overall conformation, whereas the (**d**) hydrogen bonding analysis reveals the average of 6–8 hydrogen bonds formed at the interface between the proteins during the simulation. The black and red color indicates original time series and average hydrogen bonds over (200 ns).

**Figure 13 biology-14-00261-f013:**
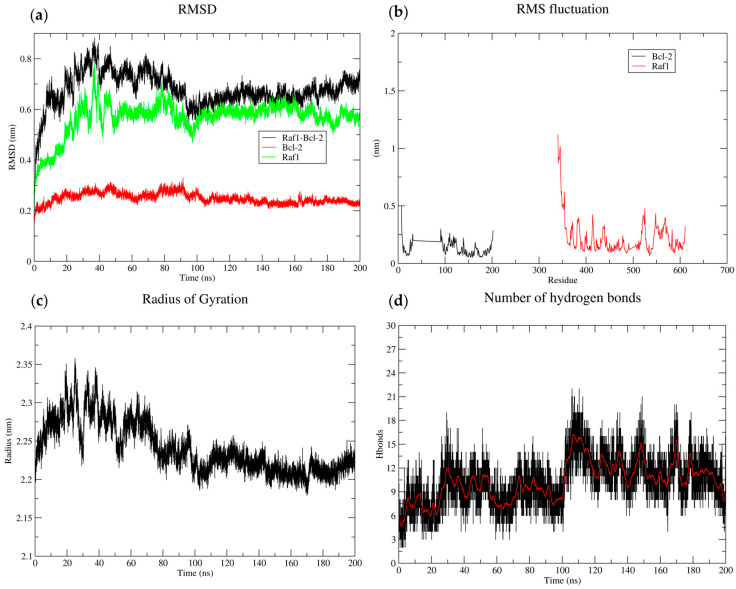
MD simulations (200 ns) of BCL2, RAF1, and BCL2-RAF1 complex using Gromacs. (**a**) The RMSD plot of the complex showed conformational changes at the beginning of simulation, with a plateau phase indicating stability over time. (**b**) The RMSF analysis highlighted the flexibility of individual residues within the complex. (**c**) The radius of gyration (Rg) indicates a stable overall conformation, whereas the (**d**) hydrogen bonding reveals the protein–protein stability and increased interactions. The black and red color indicates original time series and average hydrogen bonds over (200 ns).

**Figure 14 biology-14-00261-f014:**
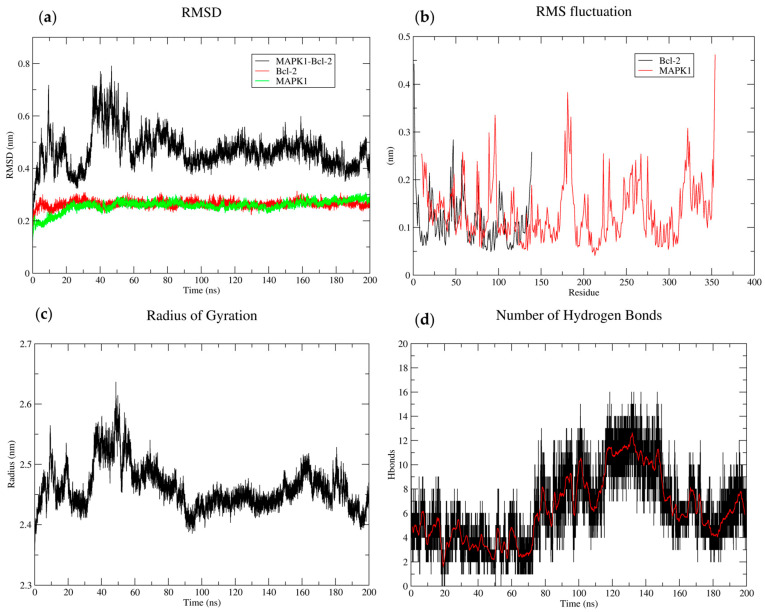
MD simulations (200 ns) of BCL2-MAPK1 complex using Gromacs. (**a**) The RMSD plot of the complex shows greater conformational changes. (**b**) The RMSF analysis highlights the flexibility of individual residues within the complex. (**c**) The radius of gyration (Rg), indicates a stable conformation, whereas the (**d**) hydrogen bonding reveals the protein–protein stability, and increased interactions. The black and red color indicates original time series and average hydrogen bonds over (200 ns).

**Table 1 biology-14-00261-t001:** BCL2 interactions with cancer drivers and cancer targets based on direct interaction.

Cancer Drivers				
Sr.	Uniprot	Gene Name	Protein Name	References
1	P04049	*RAF1*	RAF proto-oncogene serine/threonine-protein kinase	[16,17]
2	P04637	*p53*	Cellular tumor antigen p53	[18,19]
3	Q14790	*CASP8*	Caspase 8	[20,21]
4	O43521	*BCL2L11*	BCL2-like protein 11	[22,23]
5	P28482	*MAPK1*	Mitogen-activated protein kinase 1	[24,25]
6	P38398	*BRCA1*	Breast cancer type 1 susceptibility protein	[26,27]
Cancer Targets				
1	P04049	*RAF1*	RAF proto-oncogene serine/threonine-protein kinase	[16,17]
2	P98170	*XIAP*	E3 ubiquitin-protein ligase XIAP	[28,29]
3	P09874	*PARP1*	Poly [ADP-ribose] polymerase 1	[30,31]
4	P04637	*p53*	Cellular tumor antigen p53	[18,19]
5	P42574	*CASP3*	Caspase 3	[32,33]
6	P28482	*MAPK1*	Mitogen-activated protein kinase 1	[24,25]
7	P45983	*MAPK8*	Mitogen-activated protein kinase 8	[34]
8	Q5S007	*LRRK2*	Leucine-rich repeat serine/threonine-protein kinase 2	[35,36]
9	P06493	*CDK1*	Cyclin-dependent kinase 1	[37,38]

**Table 2 biology-14-00261-t002:** Key BCL2 protein partners involved in apoptosis regulation.

Name	Proteins	Structural Homology	Location	Functions	Uniprot ID
Pro-survival	BCL2	BH1-4 domains and a transmembrane domain	ER, MOM, NM	Suppresses apoptosis	P10415
	BCL-XL		ER, MOM, NM	Inhibitor of cell death	Q07817
	BCLW		MOM, cytoplasm	Promotes cell survival	Q92843
	MCL-1		Cytoplasm, MOM, nucleus	Regulation of apoptosis	Q07820
	A1		Cytoplasm	Retards apoptosis	Q16548
Effector proteins	BAX	Share homology in all four domains	Cytoplasm, MOM	Apoptosis	Q07812
	BAK		MOM, ER	Promotes apoptosis	Q16611
	BOK		Cytoplasm, ER, MOM	Apoptosis regulator	Q9UMX3
BH3-only proteins	BAD	Share homology in the BH3 only domain	Cytoplasm, MOM	Promotes cell death	Q92934
	BID		Cytoplasm, MOM	Induces apoptosis	P55957
	BIK		ER, MM, NM	Accelerates apoptosis	Q13323
	BIM		MOM, cytoskeleton	Induces apoptosis	O43521
	PUMA		Cytoplasm, MOM	Mediator of p53, induces apoptosis	Q9BXH1
	Noxa		Cytoplasm, MOM	Promotes apoptosis	Q13794
	HRK		MOM	Promotes apoptosis	O00198
	BMF		Cytoplasm, MOM	Apoptosis	Q96LC9
Tumor protein 53	p53		Cytoplasm, cytoskeleton, ER, mitochondrion, nucleus	Induces apoptosis	P04637
Serine/threonine kinase	RAF1		Cytoplasm, nucleus, mitochondrion	Retards apoptosis	P04049
Mitogen-activated protein kinase 1	MAPK1		Cytoplasm, cell junction, cytoskeleton, mitochondrion, nucleus	Induces apoptosis	P28482

ER: endoplasmic reticulum; MOM: mitochondrial outer membrane; NM: nuclear membrane.

## Data Availability

The datasets generated and/or analyzed during this study are available from the corresponding author on reasonable request.

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
