# Peer review of "Exploring the Role of BCL2 Interactome in Cancer: A Protein/Residue Interaction Network Analysis"

_biology, 2025, doi:10.3390/biology14030261_

Round 1
Reviewer 1 Report
Comments and Suggestions for Authors
The manuscript titled "Exploring the Role of BCL2 Interactome in Cancer: A Protein/Residue Interaction Network Analysis" by Sidra Ilyas and Donghun Lee focuses on BCL2 interactions with P53, MAPK1, and RAF1, offering valuable insights into the molecular mechanisms through which BCL2 regulates apoptosis.
Comments:
-
The interactions between BCL2 and P53, MAPK1, and RAF1 are well-established and widely recognized. Did the PINA study or analysis identify any new interaction partners? The current study lacks novelty The manuscript lacks a clear indication of novelty in its findings.
-
It is not understood what is the rationale for performing docking studies in this analysis? despite of availability of crystal structure of partner protein?
- Have they identified new binding sites or interacting residues in the current study
- Please provide the BCL2 interaction partner structure figure before and after docking. Is the docking site the same in both cases?
- What is the scenario for the MD simulation of PDB ID 8HLM compared to BCL2 docked with P53? Are they the same?
Author Response
Reviewer 1
The manuscript titled "Exploring the Role of BCL2 Interactome in Cancer: A Protein/Residue Interaction Network Analysis" by Sidra Ilyas and Donghun Lee focuses on BCL2 interactions with P53, MAPK1, and RAF1, offering valuable insights into the molecular mechanisms through which BCL2 regulates apoptosis.
Comments:
- The interactions between BCL2 and P53, MAPK1, and RAF1 are well-established and widely recognized. Did the PINA study or analysis identify any new interaction partners? The current study lacks novelty. The manuscript lacks a clear indication of novelty in its findings.
Thank you for your insightful comment regarding the novelty of our study. We acknowledge that the interactions between BCL2 with its partner, p53, is well documented in the literature. However, our analysis also highlighted an expanded interaction BCL2-p53 network which was absent in the X-ray structure. Additionally, the structures of BCL2 with MAPK1 and RAF1 were unavailable in PDB, the detailed interactions and their specific residue-level contributions were performed by docking studies that would provide a deeper understanding of how these proteins may interact and influence each other at a molecular level.
We believe that these insights offer a new dimension to the understanding of BCL2 interactions and their implications in cellular processes. The study’s findings emphasize conserved binding sites on BCL2 that are critical for its interactions with key partners. These insights can inform the design of therapeutic strategies to selectively target BCL2-protein interactions, a perspective that adds translational value to the research. Thus, while we acknowledge that the interactions are recognized, our study stands out by providing deeper mechanistic insights, dynamic interaction details, and therapeutic implications, contributing to the field in a meaningful way. Our findings suggest that the BCL2 interface with partner proteins may exhibit dynamic and flexible binding modes, providing new insights into its potential as a therapeutic target.
- It is not understood what is the rationale for performing docking studies in this analysis? Despite of availability of crystal structure of partner protein?
Thank you for the valuable feedback. We acknowledge that the availability of the crystal structure BCL2- p53 (8HLM) provides critical information. However, no crystal structures exist for the BCL2-RAF1 and BCL2-MAPK1. These interactions are critical to understanding the molecular basis of BCL2’s anti-apoptotic activity in cancer. Therefore, molecular docking studies were employed to predict the interactions and explore the binding dynamics of BCL2 with RAF1 and MAPK1. This approach allows us to identify potential interacting residues, conserved binding sites, and structural rearrangements that are key to their functional roles in cancer signaling pathways. We have added this detail into methods section.
- Have they identified new binding sites or interacting residues in the current study
Thanks for the valuable comment. In the current study, we analyzed the interaction between BCL-2-p53 using HDOCK to evaluate interacting residues. While the X-ray crystallographic data provides insights into key residues involved in the interaction, our docking studies has identified new interacting residues that was not previously highlighted in the crystallographic data. For example, residues such as both ALA100 and ASP103 of BCL2 interact with SER166 of p53 via hydrogen bonding in the docked structure that was absent in the X-ray data. These findings extend the understanding of BCL2-p53 interactions and may offer insights into the dynamic behavior of the complex (shown in results). Moreover, a number of unique residues that interact with both hydrogen bonds and non-hydrogen bonded contacts were also observed in BCL2-RAF1, and BCL2-MAPK1 that contributed to the overall stability and flexibility as shown in the results section.
- Please provide the BCL2 interaction partner structure figure before and after docking. Is the docking site the same in both cases?
Thanks for the feedback. We have provided the figure before and after docking. The docking site is same in both cases.
- What is the scenario for the MD simulation of PDB ID 8HLM compared to BCL2 docked with P53? Are they the same?
Thanks for the insightful comment. Yes, MD simulation and docked complexes are the same. The same simulation protocols, such as solvation models, force fields, and environmental conditions, were used to ensure consistency and comparability.
Reviewer 2 Report
Comments and Suggestions for Authors
Title: Exploring the role of BCL2 interactome in cancer: A Protein/Residue Interaction Network Analysis.
Summary
In this article, Sidra Ilyas et al. investigated the protein-protein interactions (PPIs) of BCL2 with potential binding partners and their role in cancers. Three key interactors (p53, RAF1, and MAPK1) were identified as crucial players in the cancer-related activity of BCL2 through the construction of a comprehensive PPI network and subsequent analysis using multiple bioinformatics tools (PINA, MOE, STRING, RING, and gProfiler). However, this manuscript is not publishable in Biology without revision.
Major and minor comments are listed below.
1. The sections of the manuscript should be numbered according to the journal's formatting guidelines to improve readability and consistency.
2. The protein BCL2, as the central focus of this study, should be prominently shown and highlighted in the Graphical Abstract.
3. As this is a research article, Figure 1 is unnecessary and should be removed.
Table 1 should be removed to keep the section concise and focused.
The authors should keep the Introduction section concise and coherent.
4. The Figure 2 workflow diagram adds limited value to the article. Consider removing this figure.
5. The quality of Figure 4 needs improvement as the protein names are currently difficult to identify.
6. Many BCL2 interactors were identified using bioinformatics tools in this study. However, the authors should also discuss experimentally confirmed interactors of BCL2. For example, a recent article has shown BCL2 interacting with a lysosomal ion channel (Biomolecules 2023, 13(5), 802). The authors should mention such interactors and elaborate on their biological significance in the Introduction or relevant sections.
7. The quality of Figure 5 should be improved.
8. Figure 6 does not provide significant information and should be removed.
9. Figures 7, 8, and 9: these figures address the same topic and should be combined into a single, comprehensive figure. The Figure legends are simplified if the figures are combined.
10. Figures 11, 12, and 13: the panels within these figures should be labeled (e.g., A, B, C, and D) for better interpretation.
For the Figure 12, confirm and address the observed gap in the RMS fluctuation, as this could indicate a potential issue with the data.
11. Supplementary Materials are not provided. Ensure these are included and cross-referenced within the manuscript.
Comments on the Quality of English LanguageThe English could be improved to more clearly express the research.
Author Response
Reviewer 2
Summary
In this article, Sidra Ilyas et al. investigated the protein-protein interactions (PPIs) of BCL2 with potential binding partners and their role in cancers. Three key interactors (p53, RAF1, and MAPK1) were identified as crucial players in the cancer-related activity of BCL2 through the construction of a comprehensive PPI network and subsequent analysis using multiple bioinformatics tools (PINA, MOE, STRING, RING, and gProfiler). However, this manuscript is not publishable in Biology without revision.
Major and minor comments are listed below.
- The sections of the manuscript should be numbered according to the journal's formatting guidelines to improve readability and consistency.
Thanks for the insightful comment. We have numbered the manuscript according to the journal’s requirement.
- The protein BCL2, as the central focus of this study, should be prominently shown and highlighted in the Graphical Abstract.
Thank you for your valuable comment. We have highlighted the BCL2 protein in the Graphical Abstract.
- As this is a research article, Figure 1 is unnecessary and should be removed.
Table 1 should be removed to keep the section concise and focused.
The authors should keep the Introduction section concise and coherent.
Thanks for the feedback. The figure 1 and table 1 are removed. We have kept the Introduction section concise and coherent.
- The Figure 2 workflow diagram adds limited value to the article. Consider removing this figure.
Thank you for your valuable feedback. The Figure 2 depicts the workflow used in the study, may appear straightforward, we believe it is an important addition for several reasons. The diagram provides a clear and concise visual representation of the methodology, which helps readers quickly grasp the sequence of steps involved in our research. This is particularly useful for readers who are new to the field or unfamiliar with the specific approach we employed. Additionally, it serves as a helpful reference point for understanding the flow of experiments and analyses described in the text. Keeping the figure 2 offer a more accessible and structured understanding of our research process, ensuring that key steps are easily identifiable. We believe this will enhance the overall clarity and readability of the manuscript, and we respectfully request to retain it in the final version.
- The quality of Figure 4 needs improvement as the protein names are currently difficult to identify.
Thanks for the feedback. The figure 4 is improved now
- Many BCL2 interactors were identified using bioinformatics tools in this study. However, the authors should also discuss experimentally confirmed interactors of BCL2. For example, a recent article has shown BCL2 interacting with a lysosomal ion channel (Biomolecules2023, 13(5), 802). The authors should mention such interactors and elaborate on their biological significance in the Introduction or relevant sections.
Thanks for the valuable feedback. We have added experimentally confirmed BCL2 interactors in the Introduction sections.
- The quality of Figure 5 should be improved.
Figure 5 is improved
- Figure 6 does not provide significant information and should be removed.
Thanks for the feedback. The figure 6 has been removed.
- Figures 7, 8, and 9: these figures address the same topic and should be combined into a single, comprehensive figure. The Figure legends are simplified if the figures are combined.
Thank you for your valuable feedback. While we understand the suggestion to combine Figures 7, 8, and 9 into a single comprehensive figure, we believe that keeping them separate enhances the clarity and readability of the results. Each figure highlights a specific aspect of the analysis, and combining them would make the figure overly large and visually dense, potentially reducing the visibility and interpretability of the data. We respectfully request to retain the figures in their current format, as this approach supports better visualization and comprehension of the data.
- Figures 11, 12, and 13: the panels within these figures should be labeled (e.g., A, B, C, and D) for better interpretation.
Thanks for the feedback. We have labelled the Figures 11, 12, and 13 for better interpretation
For the Figure 12, confirm and address the observed gap in the RMS fluctuation, as this could indicate a potential issue with the data.
Thank you for your comment regarding the observed gap in the RMS fluctuation plot. The gap corresponds to the sequence alignment and residue numbering differences between the two proteins used in our study. These proteins have different residue ranges, and the gap reflects the regions where residues from RAF1 do not align with BCL2. We confirm that the data is accurate, and this gap is not indicative of missing residues or errors in our analysis. Instead, it arises from the fact that the RMS fluctuation plot includes residues for both proteins sequentially, leading to the visual discontinuity in residue numbering between the two chains.
- Supplementary Materials are not provided. Ensure these are included and cross-referenced within the manuscript.
Thank you for your observation. We sincerely apologize for the oversight in uploading the Supplementary Materials file. The Supplementary Materials contain additional data, detailed and supporting analyses that are crucial to complementing the main text. We have now included the Supplementary Materials and ensured that all references to supplementary figures, tables, and methods are appropriately cross-referenced within the manuscript. We appreciate your attention to this matter and regret any inconvenience caused by its initial omission.
Round 2
Reviewer 1 Report
Comments and Suggestions for Authors
Dear authors,
Thank you for addressing some of the comments. The manuscript can be accepted once the remaining concerns are addressed..
1. Provide the figures showing the expanded contacts identified before and after the MD simulation for each complex. For instance, include a superimposed structure of the complex before and after the MD simulation, highlighting the key residues identified in the current study.
2. Mention the units for docking score and confidence score in the supplementary file Table S9? Also, the ligand RMSD for BCL2-RAF1 and BCL2-MAPK1 complexes is higher compared to the BCL2-p53 complex explain?.
3. The authors state in their response that no crystal structures exist for BCL2-RAF1 and BCL2-MAPK1. However, in line 144 of the manuscript, they reference the PDB ID for these structures. Kindly clarify/correct the statement!
4. For the BCL2-MAPK1 complex, the RMSD increased from 0.4 to 0.6 nm between 40 and 60 ns, before decreasing to approximately 0.5 nm. Please include an explanation for this trend in the manuscript.
Author Response
- Provide the figures showing the expanded contacts identified before and after the MD simulation for each complex. For instance, include a superimposed structure of the complex before and after the MD simulation, highlighting the key residues identified in the current study.
We appreciate the reviewer's suggestion to visualize the expanded contacts identified before and after MD simulation. We have generated supplementary figures (S1, S2 and S3) depicting the superimposed structures of each complex (BCL2-p53, BCL2-RAF1, and BCL2-MAPK1) before and after MD simulation. While we understand the need to highlight the key residues involved in these interactions, we found that directly highlighting all relevant residues on the superimposed structures resulted in visual clutter, making it difficult to interpret the interactions clearly. To overcome this challenge, we have included supplementary tables (S1, S2 and S3) that lists the key residues involved in the expanded contacts for each complex, both before and after MD simulation. This table provides a clear and concise overview of the identified interactions and facilitates easy comparison between the pre- and post-simulation states. We believe this approach effectively conveys the information about the expanded contacts while maintaining the clarity and visual appeal of the figures.
- Mention the units for docking score and confidence score in the supplementary file Table S9? Also, the ligand RMSD for BCL2-RAF1 and BCL2-MAPK1 complexes is higher compared to the BCL2-p53 complex explain?.
We acknowledge the reviewer's feedback. We have mentioned the docking scores in kcal/mol in the supplementary file Table S9, however; confidence score is unit less as it is a measure of how confident the docking algorithm is in the predicted binding pose.
We acknowledge the reviewer's observation that the RMSD values for BCL2-RAF1 and BCL2-MAPK1 are higher than for BCL2-p53. It is important to note that we have employed template-free docking using HDOCK, which relies solely on protein structures and does not utilize any prior information from known complexes. This approach can be more challenging, particularly for flexible proteins like RAF1 and MAPK1. The higher RMSD values for these complexes might reflect the inherent flexibility of these proteins and the difficulty in accurately predicting their binding poses without structural templates. Additionally, the binding sites on RAF1 and MAPK1 might be more complex than the binding site on p53, further complicating the docking process. While template-free docking is a powerful approach, it has limitations, and the observed higher RMSD values might reflect these challenges.
- The authors state in their response that no crystal structures exist for BCL2-RAF1 and BCL2-MAPK1. However, in line 144 of the manuscript, they reference the PDB ID for these structures. Kindly clarify/correct the statement!
We acknowledge the reviewer's feedback. These PDB IDs (3OMV and 2GPH) correspond to structures of RAF1 and MAPK1 in complex with other proteins/compounds, not with BCL2. Therefore, we have used the crystal structures of RAF1 (3OMV) and MAPK1 (2GPH) as templates for docking simulations, as no crystal structures of BCL2-RAF1 and BCL2-MAPK1 complexes are currently available. We have clarified this statement in the manuscript under the section “2.7. Protein Structures retrieval and Preparation”.
- For the BCL2-MAPK1 complex, the RMSD increased from 0.4 to 0.6 nm between 40 and 60 ns, before decreasing to approximately 0.5 nm. Please include an explanation for this trend in the manuscript.
We thanks reviewer for the valuable comment. We have mentioned this trend in the manuscript at line 395.
Reviewer 2 Report
Comments and Suggestions for Authors
-It is a serious concern that the interactions between BCL2 and three partners-p53, RAF1, and MAPK1 are not experimentally confirmed.
-The quality of figures should be improved.
Author Response
It is a serious concern that the interactions between BCL2 and three partners-p53, RAF1, and MAPK1 are not experimentally confirmed.
We acknowledge the reviewer's concern regarding the experimental confirmation of the BCL2 and three partners-p53, RAF1, and MAPK1 interaction. The interaction between the partners-p53, RAF1, and MAPK1 proteins has been experimentally supported by multiple independent studies, including affinity capture-western, two-hybrid assays, co-crystal structure, co-fractionation, co-purification, protein-peptide, reconstituted complex and biochemical activity assays. These findings are evident by data available in the BioGRID database https://thebiogrid.org/107068/summary/homo-sapiens/bcl2.html, a highly curated resource for PPI data. This evidence strongly supports the existence of a physical and functional interaction between BCL2 and three partners-p53, RAF1, and MAPK1 as mentioned in the current study.
-The quality of figures should be improved.
We appreciate the reviewer's feedback regarding figure quality. We have achieved maximum quality for all figures in the manuscript.